# Analysis of Flavonoid Metabolites in *Citrus reticulata ‘Chachi’* at Different Collection Stages Using UPLC-ESI-MS/MS

**DOI:** 10.3390/foods12213945

**Published:** 2023-10-28

**Authors:** Yuting Chen, Yujuan Xu, Jing Wen, Yuanshan Yu, Jijun Wu, Lina Cheng, Wai-San Cheang, Wenwen Liu, Manqin Fu

**Affiliations:** 1Sericultural & Argi-Food Research Institute, Guangdong Academy of Agricultural Sciences/Key Laboratory of Functional Foods, Ministry of Agriculture and Rural Affairs/Guangdong Key Laboratory of Agricultural Products Processing, No. 133 Yiheng Street, Tianhe District, Guangzhou 510610, China; ywcyt0625@163.com (Y.C.); xuyujuan@gdaas.cn (Y.X.); wenjing@gdaas.cn (J.W.); yuyuanshan@gdaas.cn (Y.Y.); wujijun@gdaas.cn (J.W.); chenglina@gdaas.cn (L.C.); 2College of Food and Pharmaceutical Engineering, Wuzhou University, Wuzhou 543002, China; 3Institute of Chinese Medical Sciences, State Key Laboratory of Quality Research in Chinese Medicine, University of Macau, Macao SAR 999078, China; annacheang@um.edu.mo; 4Institute of Quality Standard and Monitoring Technology for Agro-Products of Guangdong Academy of Agricultural Sciences, Guangzhou 510640, China; liuwenwen@gdaas.cn

**Keywords:** flavonoid metabolites, UPLC-ESI-MS/MS, *Citrus reticulata ‘Chachi’*, PCA, OPLS-DA

## Abstract

Flavonoids are essential substances with antioxidant properties and high medicinal value. *Citrus reticulata ‘Chachi’ peel* (CRCP) is rich in flavonoids and has numerous health benefits. The different maturity periods of CRCP can affect the flavonoid contents and pharmacological effects. In this study, we successfully performed UPLC-ESI-MS/MS-based metabolic analysis to compare the metabolites of CRCP at different harvesting periods (Jul, Aug, Sep, Oct, Nov, and Dec) using a systematic approach. The results revealed the identification of a total of 168 flavonoid metabolites, including 61 flavones, 54 flavonols, 14 flavone C-glycosides, 14 dihydroflavones, 9 flavanones, 8 isoflavones, 3 flavanols, 3 dihydroflavonols, and 2 chalcones. Clustering analysis and PCA were used to separate the CRCP samples collected at different stages. Furthermore, from July to December, the relative contents of isoflavones, dihydroflavones, and dihydroflavonols gradually increased and flavanols gradually decreased over time. The relative content of flavonoid C-glycosides showed an increasing and then decreasing trend, reaching the highest value in August. This study contributes to a better understanding of flavonoid metabolites in CRCP at different harvesting stages and informs their potential future utilization.

## 1. Introduction

*Citrus reticulata ‘Chachi’* is an ancient citrus variety that has been cultivated in Guangdong Xinhui for over 700 years [1]. *Citrus reticulata ‘Chachi’* peels (CRCP) serve as the authentic raw material for *Pericarpium Citri Reticulatae*, known as Chenpi in Chinese [2]. Chenpi is a traditional Chinese medicine that has been included in the Chinese Pharmacopoeia [3] and is also recognized as a “National Geographical Indication” protected product [4]. CRCP has important food and medicinal values and is mainly used for the prevention and treatment of digestive and respiratory diseases [2]. CRCP is rich in bioactive constituents, especially flavonoids, which have a wide range of health benefits. These benefits include the regulation of vital energy, anti-tumor activity, pain relief, muscle relaxation, antioxidant effects, and anti-neuroinflammatory properties [5,6,7,8]. Phytochemical and pharmacological studies demonstrated that the major components of CRCP are dietary flavonoids, which are generally categorized into two groups: flavanone glycosides (primarily hesperidin) and polymethoxylated flavones (PMFs, primarily nobiletin and tangeretin) [2]. In addition, due to its unique growing conditions, the medicinal value and total flavonoid content of CRCP is higher than that of other citrus plants [9,10].

Traditionally, *Citrus reticulata ‘Chachi’* is harvested in three stages: unripe, semi-ripe, and ripe. As the fruit grows, its color transitions from green to yellow and finally to a deep orange. These different stages of maturity have an impact on the flavonoid content and pharmacological properties [11]. According to the maturity of citrus fruits on the market, it is generally divided into green peel, two red peel, and three kinds of red peel, which are the products of different time periods during the growth period of Xinhui citrus fruits, mainly according to the maturity of the fruits, and the main effects are as follows. Green peel: green peel’s volatile oil content is relatively high, it has storage resistance, nourishes the spleen and qi, and is suitable as a medicine in herbal tea. Two red skin: two red skin’s nutrient content is evenly distributed, the effect tends to calm, it is easy to store, it can be used as the main purpose of usual medicine, and it can also be used as a general meat seasoning. Large red skin: large red skin’s characteristics are that it promotes calm, is smooth, is applicable to a wide range of contexts, it can be matched with valuable Chinese herbs or with qi to strengthen the spleen and stomach, and it targets dampness cough, phlegm, and other effects. In summary, the green skin focuses on dispersing the liver and breaking down qi, eliminating accumulation and resolving stagnation; the red skin focuses on regulating qi and strengthening the spleen, drying dampness and resolving phlegm [3,12]. The variation in appearance, nutrition, taste, and biological activity during the ripening process results in *Citrus reticulata ‘Chachi’* exhibiting diverse metabolite profiles [13]. A previous study [14] reported that the flavonoids hesperidin, chrysin, and tangerine peel may be the chemical markers responsible for differences in the ripeness of citrus fruits. The CRCP demonstrates changes in flavonoid content at different fruit picking stages. Therefore, there is a need for the systematic analysis of flavonoids and screening of unique components in the CRCP of different maturity levels.

Metabolomics analysis is a potent and useful tool, which is increasingly being utilized in plant research for identifying and comparing chemical components [15,16,17]. Metabolomics analyses were particularly effective in analyzing the chemical profiles of fruits from different origins [18,19], growing locations [20], cultivation ages [21], and processing methods [22], and they played an important role in the variation of chemical characteristics of fruit. Metabolomics has previously been employed to identify *Citri Reticulatae Pericarpium* of different storage years [3,12], investigate variations in the composition of secondary metabolites among citrus juices [23], and analyze citrus fruits of different varieties [24,25]. Metabolomics has also been used to analyze the primary metabolites of different citrus varieties, including grapefruit, sweet orange, and Persian sour orange, at different fruit picking stages and suggested the metabolic pathways associated with citrus fruit development [26,27,28,29,30]. Indeed, the UPLC-ESI-MS/MS-based analysis of flavonoids is frequently combined with the Orthogonal Partial Least Squares–Discriminant Analysis (OPLS-DA) and Principal Component Analysis (PCA). OPLS-DA is widely used in metabolomics research because of its ability to assess similarities and differences between samples and identify key discriminants that lead to differences in variables in the model [3,12]. PCA is used to reduce the dimensionality of the data to visualize the overall pattern and clustering of the data set. OPLS-DA and PCA play crucial roles in synthesizing and interpreting flavonoid data obtained through UPLC-ESI-MS/MS.

In a previous study, the physical and chemical properties and the changes in organic acids, total flavonoids, total phenols, and other active components of *Citrus reticulata ‘Chachi’* at different collection stages were studied [31]. Liu et al. [32] used headspace gas chromatography–ion mobility spectrometry (HS-GC-IMS) and partial least squares–discriminant analysis (PLS-DA) for a rapid and comprehensive evaluation of flavor compounds in dried fruits and identified 86 in the harvested period. In this study, for the first time, UPLC-MS was used to analyze the changes in flavonoids in CRCP samples collected from six different fruit harvesting periods from July to December. In order to comprehensively assess the dynamic changes in multiple components, the Variable Importance in Projection (VIP) method, based on the PCA and OPLS-DA models, was used to analyze the flavonoid components. The results of this study have the potential to provide a theoretical basis for the scientific harvesting, quality control, and further processing of CRCP.

## 2. Materials and Methods

### 2.1. Plant Materials

The samples used in this study were from the Citrus Professional Cooperative of Shuangshui Town, Xinhui District, Jiangmen City, Guangdong Province, China (23.05° N, 112.46° E). The sampling process consisted of selecting five fruit trees from each of the five designated sampling sites in the east, south, west, north, and center of the farm. Sampling took place on the 20th of each month from July to December 2021. Only fresh fruits of uniform shape, medium size, smooth surface, and free from moth infection were selected for sampling. On average, about 1 kg of fruit was collected from each tree. The collected samples were washed, sterilized, peeled, and dried to obtain CRCP, and then the dried CRCP samples were pulverized into a fine powder by passing the samples through a 100 mesh sieve to obtain sample powders ranging in size from 0.104 mm to 0.149 mm. The samples obtained were stored at 4 °C for further analysis.

### 2.2. Sample Preparation and Extraction

The CRCP dried powders from different harvesting stages were freeze-dried using a vacuum freeze-dryer (Scientz-100F, Shanghai Yetuo Technology Co., Ltd. Shanghai, China). The freeze-dried samples were then pulverized using a zirconia bead mixer-mill (MM 400, Retsch, Scientific Industries, Bohemia, NY, USA) at 30 Hz for 1.5 min. Then, 100 mg of powder was weighed and extracted with 1.2 mL of 70% methanol solution. The extracted mixture was vortexed for 30 s at 30 min intervals and the process was repeated six times. The samples were placed in a refrigerator at 4 °C overnight. After centrifugation at 12,000× *g* for 10 min, the supernatant was filtered through a membrane filter (SCAA-104, ANPEL, Shanghai, China, http://www.anpel.com.cn/) with a pore size of 0.22 μm and then analyzed using UPLC-MS/MS.

### 2.3. UPLC Conditions

Sample extracts were analyzed using LC-ESI-MS/MS system (UPLC, Shim-pack UFLC SHIMADZU CBM A system, AB SCIEX, Framingham, MA, USA; MS, QTRAP^®^ 4500+ system, AB SCIEX, Framingham, USA). UPLC conditions were performed as previously reported [33].

UHPLC analytical conditions were as follows: column, Waters ACQUITY UPLC HSS T3 C18 (1.8 µm, 2.1 mm × 100 mm); column temperature, 40 °C; flow rate, 0.4 mL/min; injection volume, 2 μL; solvent system, water (0.1% formic acid): acetonitrile (0.1% formic acid); gradient program at 0 min, 95:5 *v*/*v*, at 10.0 min, 5:95 *v*/*v*, at 11.0 min, 5:95 *v*/*v*, at 11.1 min, 95:5 *v*/*v*, and at 15.0 min, 95:5 *v*/*v*. Both formic acid and acetonitrile were obtained from thermo Fisher Scientific (Waltham, MA, USA)

### 2.4. ESI-Q TRAP-MS/MS

Mass spectrometry analysis was performed as previously described (Chen et al., 2013). LIT and triple quadrupole (QQQ) scans were obtained using a triple quadrupole linear ion trap mass spectrometer (Q TRAP) on an AB4500 Q TRAP UPLC/MS/MS system equipped with an ESI turbo-ion spray interface operating in both positive and negative ion modes and controlled using the Analyst 1.6.3 software (AB Sciex). The ESI source The operating parameters are as follows: Ion Source, Turbo Spray; Source Temperature, 550 °C; Ion Spray Voltage (IS), 5500 V (Positive Ion Mode)/−4500 V (Negative Ion Mode); Ion Source Gases I (GSI), Gases II (GSII), and Curtain Gases (CUR) are set to 50, 60, and 25.0 psi, respectively; and Collision Activated Dissociation (CAD), High. Instrument tuning and mass calibration were performed using 10 and 100 μmol/L polypropylene glycol solutions in QQQ and LIT modes, respectively. QQQ scans were obtained as multiple reaction monitoring (MRM) experiments with the collision gas (nitrogen) set to medium. For each MRM transition, the depolymerization potential (DP) and collision energy (CE) were optimized. Further optimizations were performed for DP and CE. A specific set of MRM transitions was monitored for each period based on the elution profile of the metabolites.

### 2.5. Sample Quality Control

The secondary spectral information is used to characterize the substance on the basis of a self-constructed database. The analysis involves the removal of the following signals: isotopic signals, repetitive signals containing K+, Na+, and NH4+ ions, and repetitive signals that are themselves fragment ions of other larger molecular weight substances. In the MRM mode, precursor ions (parent ions) of the target substance are first screened using a quadrupole to exclude ions corresponding to other molecular weight substances to initially eliminate interference. The precursor ion is then induced to ionize in a collision chamber, causing it to break up and form many fragment ions. The resulting fragment ions are filtered through a triple quadrupole to select the desired one characteristic fragment ion, thus eliminating interference from non-target ions and making quantification more accurate and reproducible. After obtaining the metabolite spectral analysis data of different samples, the mass spectral peak areas of all substances are integrated and the integration is corrected for the mass spectral peaks of the same metabolite in different samples. The metabolites in the samples were characterized and quantified using mass spectrometry based on local metabolic databases. The characteristic ions associated with each metabolite were screened using triple quadrupole. The signal intensity of the characteristic ions was measured using a detector in the form of counts per second (CPS). Offline mass spectrometry files of the samples were opened using Multia Quant software (Analyst 1.6.3, AB SCIEX, Framingham, USA) which facilitates peak integration and correction. The peak area (Area) of each peak represents the relative amount of the corresponding substance in the sample. All chromatographic peak area integration data were exported and saved. Analytical 1.6.3 software (AB SCIEX AB SCIEX, Framingham, USA) was used to process the mass spectrometry data.

### 2.6. Statistical Analysis

Each experiment had three biological replicates. Cluster analysis, PCA, and OPLS-DA were performed as previously described using R (http://www.r-project.org/) [34].

## 3. Results and Discussion

### 3.1. Quality Control (QC) Analysis of Samples

Figure 1 shows the total ion current (TIC) plots of mass spectra obtained in positive and negative ion detection modes for quality control samples examined using the UPLC-MS/MS technique. The flavonoid metabolites in the peel of the nettle ‘*Chachi*’ at different harvesting stages were studied using the UPLC-MS/MS technique and database. The results showed that the TIC profiles of the metabolites were highly overlapping, with consistent retention times (RT) and peak intensities between the QC samples. The high overlap of the spectra indicates that the assay has good signal stability and provides reliable data results.

### 3.2. Classification of Flavonoid Metabolites

A total of 168 flavonoid metabolites (numbered **1**–**168**) were identified in CRCP at different collection stages. These metabolites were categorized into different classes including 61 flavones, 54 flavonols, 14 flavonoid *C*-glycosides, 14 dihydroflavonoids, 9 flavanones, 8 isoflavonoids, 3 flavanols, 3 dihydroflavonoids, and 2 chalcones, as shown in Figure 2 and Appendix A. The flavones were analyzed and found to be the most abundant among the identified metabolites in all the samples at different maturity stages, accounting for 36.31% of the total metabolites in all the samples. From July to December, there were some trends in the relative content of specific metabolite categories. The relative contents of isoflavones, dihydroflavones, and dihydroflavonols gradually increased and flavanols gradually decreased over time. On the other hand, the relative contents of flavonoid *C*-glycosides showed a pattern of increasing and then decreasing, with the highest content in August. Figure 3 shows the thermogram of all flavonoid metabolites in the homogenized samples. The flavonoid metabolite levels varied considerably in December and November compared to July, August, September, and October, whereas the flavonoid metabolite levels were consistent in September and October. Meanwhile, cluster analysis of all flavonoid metabolites revealed that more than half of the flavonoid metabolites were found to be lower in December and November compared to July, August, September, and October. The heat map and cluster analysis showed significant changes in flavonoid metabolite content in December and November compared to earlier months (July–October). This finding suggests that the composition and content of flavonoid metabolites changed dynamically over time, with December and November showing unique patterns compared to other months.

### 3.3. The Variation of Flavonoid Metabolites with High Relative Content

Heatmaps are an intuitive visualization method for analyzing the distribution of experimental data to observe patterns and trends [24,25]. To visually show the dynamic change process of each differential metabolite with developmental stages, a heatmap was made for the ionic intensity of 12 differential metabolites with high content in different picking stages (Figure 4). The contents of compound **116** vitexin, **124** genistein-8-C-glucoside, **5** velutin, **4** tricin, and **39** 5-hydroxy-3,6,7,4′-tetramethoxyflavone showed an increasing and then decreasing trend. The content of **116** vitexin and **124** genistein-8-C-glucoside reached their maximum values in August; whilst **5** velutin and **4** tricin exhibited their highest content in September. It is noteworthy that a previous study reported a similar trend of increasing and then decreasing total flavonoid content [31]. The content of **39** 5-Hydroxy-3,6,7,4′-tetramethoxyflavone was found to be the highest in October. The content of compound **79** quercetin-3-O-robinobioside, **80** rutin, **118** isoorientin, and **119** orientin showed a more complex trend of first increasing and then decreasing, followed by another increase. The content of compound **131** neohesperidin and **160** hesperdin showed a gradual upward trend throughout the different picking stages. In contrast, compound **138** natsudaidain showed a downward trend in CRCP during different picking stages. The flavonoid metabolites in CRCP may contribute to potential health benefits. Matsui et al. reported that natsudaidain has the potential to alleviate inflammatory diseases such as advanced type I allergy and chronic inflammation associated with fibrosis [35]. Hesperidin, the main dihydroflavonoid in CRCP, has been widely studied for its therapeutic effects in various diseases due to its anti-inflammatory, antioxidant, hypolipidemic, and insulin-sensitizing properties [36]. In conclusion, these findings provide insights into the dynamics of the specific compound contents at different harvesting stages of CRCP. The aforementioned compounds exhibit diverse patterns of fluctuation, with some showing multiple peaks and others displaying consistent increases or decreases.

### 3.4. Differential Flavonoid Metabolite Analysis Based on PCA and OPLS-DA

In order to determine the trends of flavonoids in CRCP samples at different harvest stages, the peak intensities of six harvest stages measured using UPLC−ESI−MS were integrated using a PCA and OPLS-DA model. A PCA is an unsupervised multivariate statistical method of pattern recognition for highlighting the specific samples in all data. In the PCA score plot, if the points of several samples are clustered together, the similarity between these samples is very high; on the contrary, if the points of several samples are very dispersed, the similarity between these samples is relatively low. In the PCA score plot, the samples collected in July, August, September, October, November, and December are separated from each other, which indicates that the differentiation between the groups (July–December) is good. The scattered points corresponding to the six samples show clustering with each other within the group, while the three parallel samples collected in different time periods are tightly clustered, which suggests that the replication within the group is better, and the data of the samples between the parallels are very similar (Figure 5). In this study, the three extracted principal components, PC1, PC2, and PC3, accounted for 43.15%, 20.47%, and 12.66%, respectively.

While PCA cannot assign the class membership of unknown test samples, OPLS-DA focuses on grouping and on the differences between grouped samples, and has better classification and prediction capacity. The trends of flavonoids in CRCP (July–December) at different stages of maturity were explored using the VIP method in the OPLS-DA model, as shown in Figure 5B. As shown in Figure 6A, the CRCP samples from different growth periods were completely separated in the OPLS−DA scoring plot, indicating that there were significant differences in the chemical properties of CRCP from different maturation periods. In addition to the score plots, the OPLS-DA analysis also yields S-plot plots (Figure 6B), where the horizontal coordinate indicates the covariance of the principal component with the metabolite and the vertical coordinate indicates the correlation coefficient of the principal component with the metabolite. The S-plot plots are generally used to select metabolites that have a strong correlation with the main components of the orthogonal signal correction (OSC) process, and on the other hand, at the same time, they can also be used to select metabolites that have a strong correlation with the Y metabolites. The metabolites closer to the two corners are more important, and the red points in the S-plot indicate that the VIP value of these metabolites is greater than or equal to 1, while the green points indicate that the VIP value of these metabolites is less than or equal to 1. The Y variable in the model represented the CRCP samples harvested at six stages, and the X variable included 168 flavonoids identified using UPLC-ESI-MS. The goodness-of-fit (R^2^X = 0.914 and R^2^Y = 0.985), predictability (Q^2^ = 0.911), and cross-validation of the OPLS-DA model, as determined using a permutation test (*n* = 200), indicated that the model was statistically acceptable (Figure 6C). The detected R^2^ and Q^2^ values were higher than 0.91, which indicates that the model is highly accurate and stable, and reliable in explaining variations in the metabolite profiles of the CRCP samples [37]. In this model, all samples from the six collection stages were well separated with no overlap, indicating that the UPLC-ESI-MS assay for flavonoids was very reliable in classifying CRCP from these different collection stages. Samples picked from August through December were distributed in the upper part of the score plot, while samples picked in July were distributed in the lower part of the score plot. These results suggest that the CRCP samples harvested from August to December had similar flavonoid compounds, while the samples harvested in July had unique flavonoid metabolite profiles. The similar flavonoid metabolite patterns of the samples harvested from August to December imply that there may be common metabolic pathways or a range of environmental factors influencing the composition of flavonoids during this period. On the other hand, the unique profile of the flavonoid metabolites in the samples picked in July suggests a distinctive metabolic profile associated with the early stages of fruit development. These findings are consistent with the results of PCA and confirm that OPLS-DA is able to effectively differentiate CRCP samples at different maturity levels on the basis of flavonoid metabolites.

### 3.5. Variation Trend of Differential Metabolites

Based on the results of OPLS-DA, the VIP values from the multivariate analysis of the OPLS-DA model were used to initially screen out the differential metabolites of CRCP at different collection stages. A total of 21 metabolites with VIP ≥ 1.5 and *p*-value ≤ 0.05 were identified. A line graph was used to show the variation of differential metabolites in the samples (Figure 7). The samples were categorized into six major groups, consistent with the results of the PCA. The contents of 14 compounds (**1**, **2**, **7**, **24**, **41**, **50**, **58**, **74**, **83**, **139**, **140**, **144**) showed a tendency of increasing and then decreasing, and reached the maximum value in September. The contents of **65** kaempferol 3-O-glucoside, **71** di-O-methylquercetin, **90** quercetin-3-O-galactoside, and **104** quercetin-7-O-(6′-O-malonyl)-β-D-glucoside increased gradually from Jul to Aug, decreased from Aug to Nov, and then increased. The contents of **148** isosakuranetin and **158** hesperetin increased gradually from Jul to Sep, decreased from Sep to Nov, and then increased. The compound **158** hesperidin has many medicinal and commercial values, among which the main functions reported include antibacterial, anti-inflammatory, antioxidant, antiviral, anti-hypertensive, anti-atherosclerosis, immune enhancement, and anti-tumor activities, and so on [38]. For the content of **151** naringenin-7-O-glucoside, the trend of first decreasing and then increasing appeared again, and it reached the maximum value in Oct. The level of **165** eriodictyol O-malonylhexoside decreased and then increased, with the highest levels in July. Isoflavones are usually found in legumes such as soybean and play an important role in plant defense and nodulation. In the present study, eight isoflavones were detected in CRCP, suggesting that isoflavones may also be present in non-leguminous plants. Among the detected isoflavones, the relative amount of **142** genistein-7-O-Glucoside was the highest, followed by prunetin, 5,7,3′,4′-tetrahydroxyisoflavone and biochanin A. Prunetin is usually found in herbs and spices such as Prunus species or Glycyrrhiza glabra [39]. Biochanin A is an isoflavone derivative isolated from red clover (Trifolium pratense) with estrogen-like, cholesterol-lowering, antifungal, and antitumor properties [40].

### 3.6. Differential Metabolite Enrichment Analysis

Flavonoids include many metabolites with different functions, some of which vary considerably at different acquisition stages. Through the functional annotation and enrichment analysis of the differential metabolites, we found that the differential metabolites were mainly involved in biosynthetic pathways. The differential flavonoid metabolites of each comparison group were annotated by searching the Kyoto Encyclopedia of Genomes (KEGG) database (https://www.kegg.jp/) to obtain detailed pathway information. In this study, we mapped 168 differential metabolites to the KEGG database. First, we focused on metabolic pathway information and found that most of the metabolites were mapped to “metabolites”. A few metabolites belonged to other system information categories, such as “genetic information processing” and “environmental information processing”. The above labeling results were enriched according to the pathway types in KEGG, and the enrichment results are shown in the bubble diagram in Figure 8. A total of 68 pathways were involved in the differential metabolite analysis in July and December, among which the top five pathways with P-values in the metabolic pathway enrichment analysis were “metabolic pathway”, “biosynthesis of secondary metabolites”, “biosynthesis of flavonoids”, “metabolism of phenylalanine”, and “metabolism of tryptophan”. Among these metabolic pathways, “biosynthesis of secondary metabolites” and “metabolic pathways” identified more differential metabolites in the July and December comparisons compared to the other metabolic pathways.

## 4. Conclusions

In this study, we successfully performed UPLC-ESI-MS/MS-based metabolic analysis to compare the metabolites of CRCP at different harvesting periods (Jul, Aug, Sep, Oct, Nov, and Dec) using a systematic approach. The results showed that the samples contained a total of 168 flavonoids at six different fruit harvesting periods. Differential metabolite analysis showed that flavones were the most dominant metabolites, accounting for 36.31% of the total. The relative contents of isoflavones, dihydroflavonoids, and dihydroflavonols gradually increased and flavanols gradually decreased over time from July to December. Eight isoflavones were detected in the CRCP, with genistein-7-O-glucoside having the highest relative content, followed by prunetin, 5,7,3′,4′-tetrahydroxyisoflavone, and biochanin A. The results of the OPLS-DA showed that the flavonoid metabolites in the samples harvested at different stages of the process were well separated from each other and could be easily differentiated. The VIP scores identified 21 representative flavonoid metabolites in the samples harvested at different stages and 21 representative flavonoid compounds. Overall, this study contributes to our understanding of the composition of CRCP flavonoid metabolites and provides valuable information for future medical use and for the development of standardized products based on CRCP.

## Figures and Tables

**Figure 1 foods-12-03945-f001:**
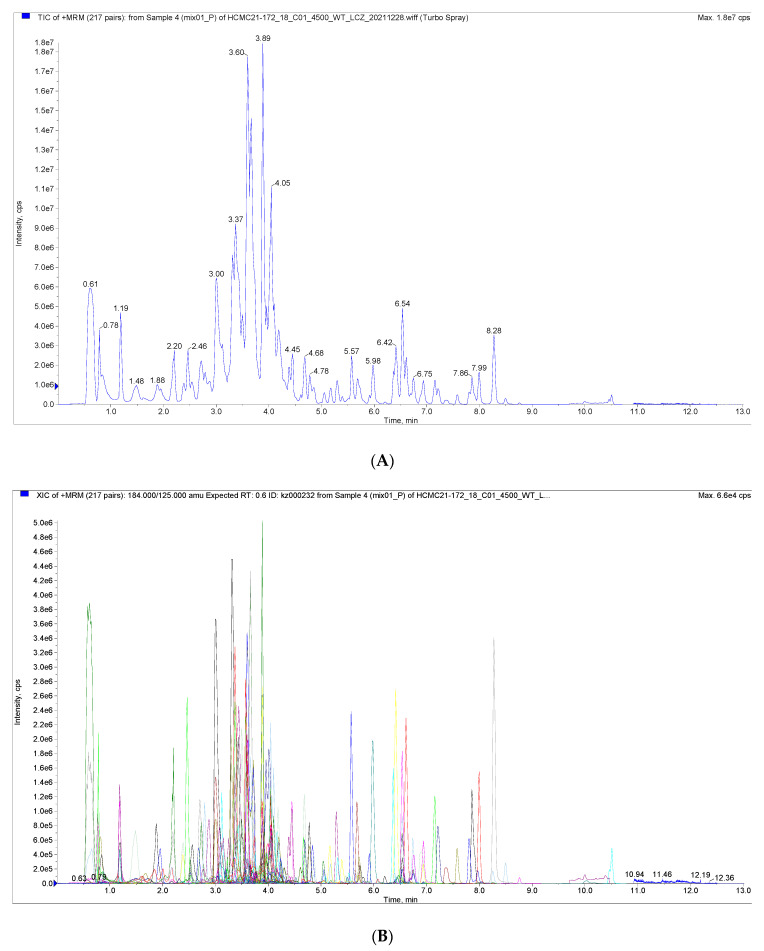
(**A**) Total Ions Current (TIC) of the mixed sample. (**B**) Overlapping of total ions flow diagram of sample.

**Figure 2 foods-12-03945-f002:**
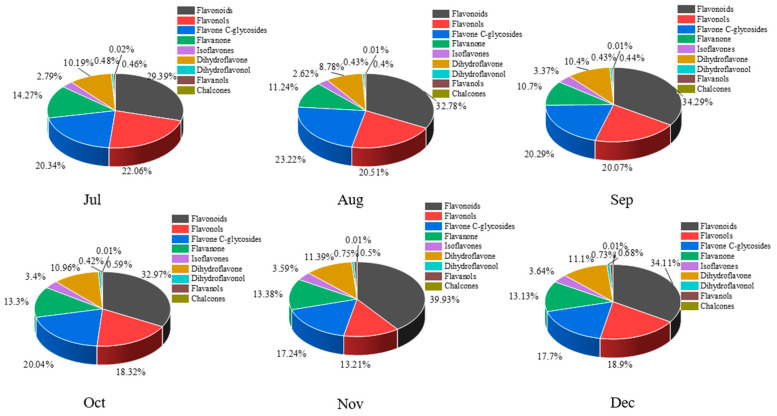
The proportion all flavonoid metabolites in CRCP at different maturity periods (July–December).

**Figure 3 foods-12-03945-f003:**
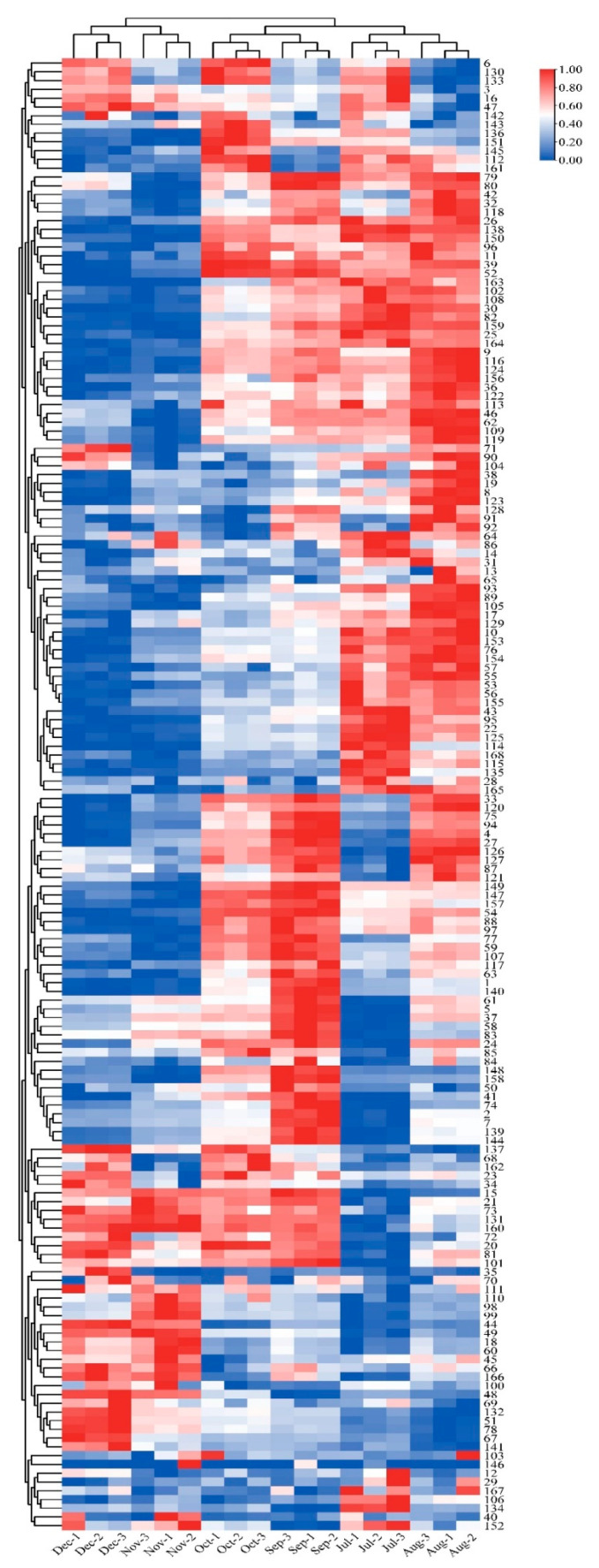
Clustering heat map of all flavonoid metabolites in CRCP at different maturity periods (July–December).

**Figure 4 foods-12-03945-f004:**
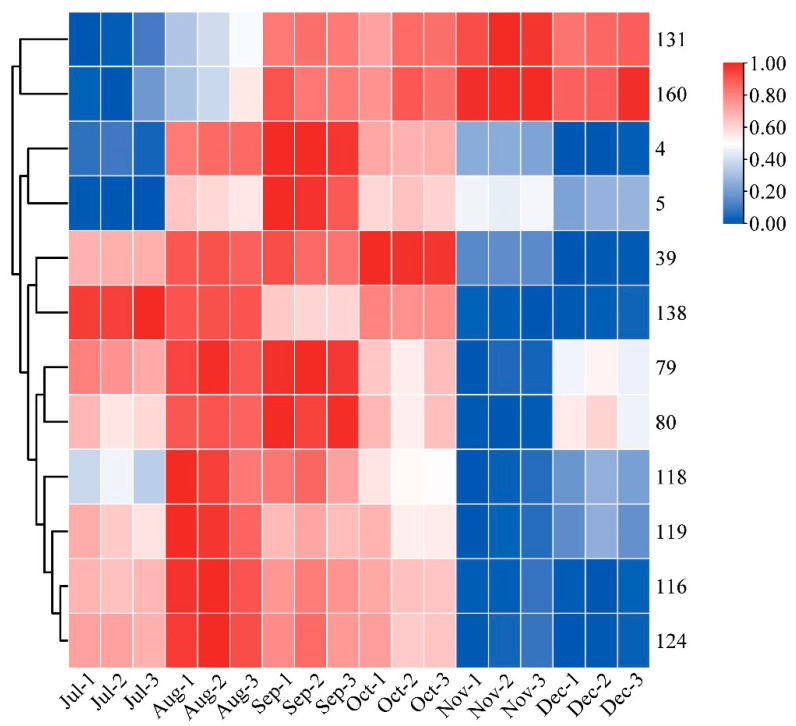
Clustering heat map of 12 flavonoid metabolites with high relative content.

**Figure 5 foods-12-03945-f005:**
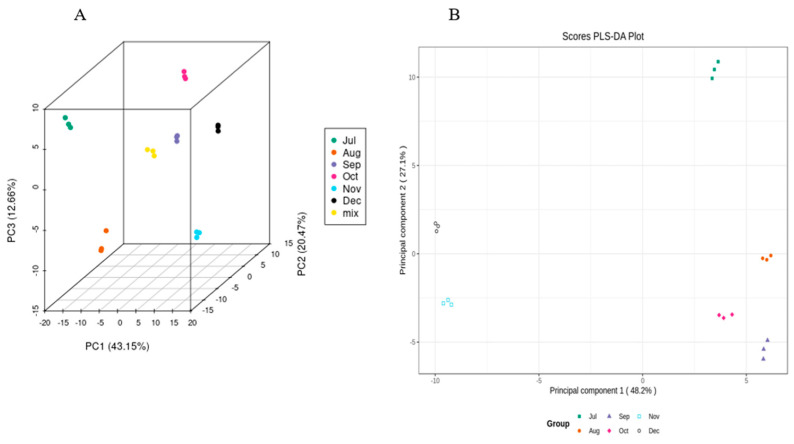
(**A**) PCA score 2D and (**B**) 3D graph of samples.

**Figure 6 foods-12-03945-f006:**
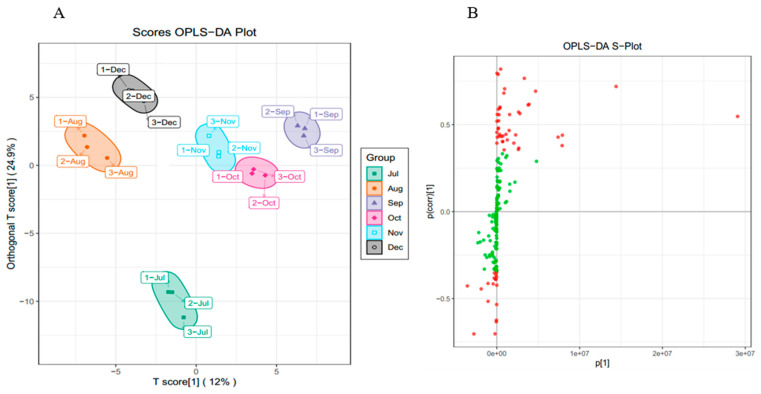
(**A**) OPLS−DA score 2D graph of samples; (**B**) OPLS−DA S−plot graph of samples; (**C**) Cross-validation plot of OPLS-DA model with 200 permutation tests.

**Figure 7 foods-12-03945-f007:**
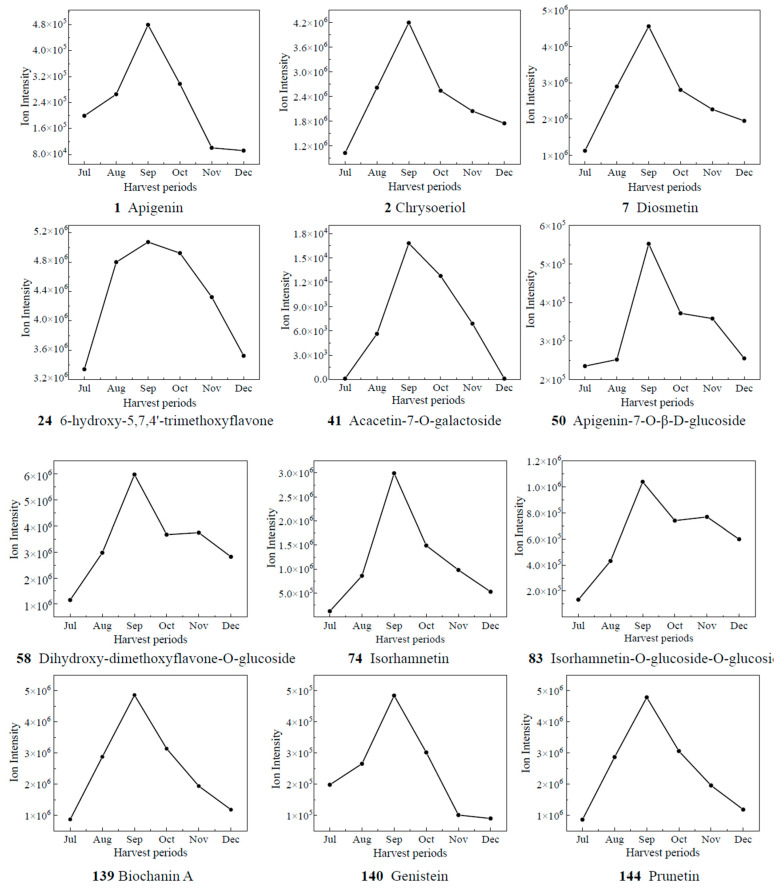
The relative contents of biomarkers of different picking stages of CRCP.

**Figure 8 foods-12-03945-f008:**
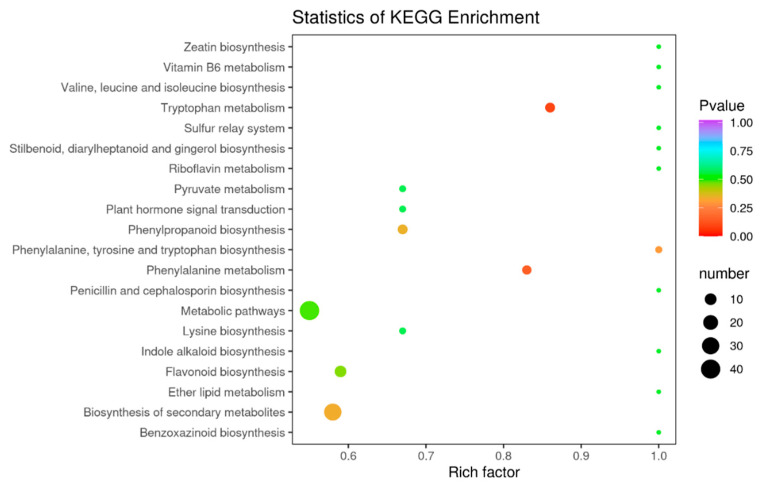
KEGG enrichment map of differential metabolites, the abscissa indicates the rich factors corresponding to each pathway, the ordinate is the pathway name, and the color of the point is the *p*-value. The redder the point, the more significant the enrichment. The sizes of the points represent the number of differential metabolites enriched.

## Data Availability

The data used to support the findings of this study can be made available by the corresponding author upon request.

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
