# Peer review of "Analysis of Flavonoid Metabolites in Citrus reticulata ‘Chachi’ at Different Collection Stages Using UPLC-ESI-MS/MS"

_foods, 2023, doi:10.3390/foods12213945_

Round 1

Reviewer 1 Report

The manuscript “Analysis of Flavonoid Metabolites in Citrus reticulata ‘chachi’ at Different Collection Stages Using UPLC-ESI-MS/MS” provides a valuable contribution in understanding flavonoid metabolites in Citrus reticulata at different ripening stages. The study is accurate in the analytical part but needs to be revised for the part containing the multivariate statistical analysis of the data.

The following changes are required.

Lines 43-45: What stage of ripeness must the fruit be in to obtain these beneficial health effects?

Line 51: It would be useful if the authors included photos of the fruit for each ripening stage

Line 77: Put full names before acronyms.

Lines 87-90: The verb is missing, rewrite the sentence.

Line 108: Enter the size of the powder obtained.

Line 258: It is not clear why to measure the repeated samples. These are considered in the PCA (in my opinion they should be eliminated from the analysis), but not in the OPLS-DA.

Lines 258-259: We do not agree with this statement. The authors should explain the concept better.

Lines 259-260: The PCA shows that the samples are different for each period, however no trend or pattern can be seen. Why? What are the variables that most impact PC1, PC2 and PC3? Without further explanation the PCA is of little significance.

Representing the results on a biplot with approximately 35% of the variability is incorrect. And all the remaining 65%? All the considerations made are therefore incorrect.

Figure 5: Figure 5C is not explained in the text, therefore not important and should be eliminated.

Author Response

Response to the Reviewer #1

  1. the manuscript “Analysis of Flavonoid Metabolites in Citrus reticulata ‘chachi’ at Different Collection Stages Using UPLC-ESI-MS/MS” provides a valuable contribution in understanding flavonoid metabolites in Citrus reticulata at different ripening stages. The study is accurate in the analytical part but needs to be revised for the part containing the multivariate statistical analysis of the data. The following changes are required.

Response: We appreciated for your suggestions and comments.

  1. Lines 43-45: What stage of ripeness must the fruit be in to obtain these beneficial health effects?

Response: We thank for the reviewer’s comment. According to the maturity of citrus fruits in the market, it is generally divided into green peel, two red peel and three kinds of red peel, which are the products of different time periods during the growth period of Xinhui citrus fruits, mainly according to the maturity of the fruits, and the main effects are as follows:

(1) Green peel: green peel volatile oil content is relatively high, storage resistance, nourish the spleen and qi, suitable as medicine pot herbal tea.

(2) Two red peel: two red skin nutrient content throughout the proportion, the effect tends to calm, easy to store, can be used as the main purpose of the usual medicine, can also be used as a general meat seasoning.

(3) Large red peel: large red skin characteristics of calm, smooth, applicable to a wide range of high appreciation; can be matched with valuable Chinese herbs, with qi to strengthen the spleen and stomach, dampness cough phlegm and other effects.

In summary, the green skin focuses on dispersing the liver and breaking down qi, eliminating accumulation and resolving stagnation; the red skin focuses on regulating qi and strengthening the spleen, drying dampness and resolving phlegm.

Reference:

Yi, L., Dong, N., Liu, S., Yi, Z., & Zhang, Y. (2015). Chemical features of Pericarpium Citri Reticulatae and Pericarpium Citri Reticulatae Viride revealed by GC–MS metabolomics analysis. Food Chemistry, 186, 192-199.

  1. Line 51: It would be useful if the authors included photos of the fruit for each ripening stage

Response: Thank you for your suggestions and comments. In this literature, we added photographs of CRCP fruits at different stages; therefore, photographs were not provided in this study.

Reference:

Chen Yuting, Fu Manqin, WU Jijun, Yu Yuan shan, Wen Jing, & Xu Yujuan. Analysis of fruit quality in different growth periods. Food and Fermentation Industries, 1-11.

  1. Line 77: Put full names before acronyms.

Response: According to the reviewer’s comment, we have revised the line 93-95 (Indeed, UPLC-ESI-MS/MS based analysis of flavonoids is frequently combined with the Orthogonal PartialLeast Squares-DiscriminantAnalysis (OPLS-DA) and Principal Component Analysis (PCA).) in the revised manuscript.

  1. Lines 87-90: The verb is missing, rewrite the sentence.

Response: According to the reviewer’s comment, we have revised the line 103-106 (H. Liu et al., 2022 used headspace gas chromatography-ion mobility spectrometry (HS-GC-IMS) and partial least squares discriminant analysis (PLS-DA) for a rapid and comprehensive evaluation of flavor compounds in dried fruits and identified 86 harvested period.) in the revised manuscript.

  1. Line 108: Enter the size of the powder obtained.

Response: According to the reviewer’s comment, we have revised the line 1224

(The collected samples were washed, sterilized, peeled and dried to obtain CRCP, and then the dried CRCP samples were pulverized into a fine powder by passing the samples through a 100-mesh sieve to obtain sample powders ranging in size from 0.104 mm to 0.149 mm.) in the revised manuscript.

  1. Line 258: It is not clear why to measure the repeated samples. These are considered in the PCA (in my opinion they should be eliminated from the analysis), but not in the OPLS-DA.

Response: I apologize for the confusion, there are no duplicate samples in either the PCA or OPLS-DA analyses, it was just an inaccuracy in line 258, where each parallel sample from a different growth period was described as a duplicate sample, I apologize for that and have been working on revising line 277-280 (In the PCA score plot, the samples collected in July, August, September, October, November, and December are separated from each other, which indicates that the differentiation between the groups (Jul-Dec) is good.) in the revised manuscript.

  1. Lines 258-259: We do not agree with this statement. The authors should explain the concept better.

Response: According to the reviewer’s comment, we have revised the line 297-298 (Trends of flavonoids in CRCP (Jul-Dec) at different stages of maturity were explored using the VIP method in the OPLS-DA model.) in the revised manuscript.

  1. Lines 259-260: The PCA shows that the samples are different for each period, however no trend or pattern can be seen. Why? What are the variables that most impact PC1, PC2 and PC3? Without further explanation the PCA is of little significance.

Response: We thank for the reviewer’s comment. In the PCA scatterplot, if the points of several samples are clustered together, it means that the similarity between these samples is very high; on the contrary, if the points of several samples are very scattered, it means that the similarity between these samples is relatively low. In this study, the scattered points corresponding to the six samples showed clustering with each other within the group, which indicates that the repeatability within the group is good and the sample data are very similar, while the differentiation between the groups (Jul-Dec) is good. Therefore, after PCA processing, we can both visualize the characteristics of each sample and cluster the samples to see the correlation and difference between the samples. Therefore, PCA analysis is still very necessary in this study.

  1. Representing the results on a biplot with approximately 35% of the variability is incorrect. And all the remaining 65%? All the considerations made are therefore incorrect.

Response: We appreciated for your suggestions and comments. I don't quite understand which biplot you are referring to, so I understand it to be Figure 7. In the manuscript, we chose flavonoids with changing trends to reveal the changes in flavonoid metabolism in the peel of CRCP at different ripening periods, and due to the word limit of the article, the flavonoids with no changing trends were not mentioned. I think this approach should be correct.

  1. Figure 5: Figure 5C is not explained in the text, therefore not important and should be eliminated.

Response: According to the reviewer’s comment, we have eliminated Figure 5C in the revised manuscript.

C

Figure 5. (A) PCA score 3D graph of samples; (B) OPLS-DA score 3D graph of samples; (C) cross-validation plot of OPLS-DA model with 200 permutation tests.

Reviewer 2 Report

The manuscript deals with the evaluation of the presence and the relative content of different metabolites in CRCP obtained from six different collection stages, and its comparison. To this purpose, UPLC-ESI-MS/MS and other data analysis were used, reaching a very descriptive and visual exposition of the results. After this, the main differences between samples from different stages were described, helping to understand better the changes occurring in CRCP composition with time. Hence, the following minor corrections should be taken under consideration:

Introduction:

-        Check: Maintain “Citrus reticulata” in italic throughout all the manuscript.

-        Line 43: Define main flavonoids found in CRCP according to literature.

-        Line 71-74: Please rephrase the sentence in order to a better understanding.

-        Line 80-83: Give a reference for the stated sentence.

-        Line 83-86: What analytical technique was used to evaluate these chemical properties and composition?

-        Line 87-90: Check this sentence, its meaning and sense is not clear.

Materials and Methods:

-        Line 107: What does “every third sample was passed through a 100-mesh sieve” mean? Was not the same procedure for all the samples?

-        Line 147: Check the title “statistical analysis”, it is the same than in subsection 2.6. Change.

Results and Discussion:

-        Figure 1. (C): If not all the calibration diagrams used for quantitative analysis and, also, the names of the metabolites are given here, this graph does not make so much sense. Add the information or, contrary, move to supplementary material.

-        Figure 2: Give more complete information in the title about the samples analyzed: for example, that they were CRCP collected from July to December… Also, for comparison between samples, it is a key factor of the research process to perform an statistical significance test in order to evaluate if differences found between samples are significant or not. Consider to include it.

-        Table S1: Consider that the total relative content for all the classes of identified compounds would give key information for the study in order to compare the differences between stages.

-        Line 198: Specify where flavonoids accounted for 36.31%.

-        Line 200-201: It is said that isoflavones, dihydroflavones, dihydroflavonols and flavanols gradually decreased over time but, observing Figure 2 it does not seem to be like that, even showing an increase over time.

-        Figure 3: As said for Figure 2, give a more detailed title.

Author Response

Response to the Reviewer #2

  1. Comments and Suggestions for Authors The manuscript deals with the evaluation of the presence and the relative content of different metabolites in CRCP obtained from six different collection stages, and its comparison. To this purpose, UPLC-ESI-MS/MS and other data analysis were used, reaching a very descriptive and visual exposition of the results. After this, the main differences between samples from different stages were described, helping to understand better the changes occurring in CRCP composition with time. Hence, the following minor corrections should be taken under consideration:

Response: We appreciated for your suggestions and comments.

  1. Check: Maintain “Citrus reticulata” in italic throughout all the manuscript.

Response: We appreciated for your advices and comments. We have checked and revised in the revised manuscript.

  1. Line 43: Define main flavonoids found in CRCP according to literature.

Response: We appreciated for your advices and comments. Phytochemical and pharmacological studies demonstrated that the major components of PCR are dietary flavonoids, which are generally categorized into two groups: flavanone glycosides (primarily hesperidin) and polymethoxylated flavones (PMFs, primarily nobiletin and tangeretin (Fu,et al,2017 ).

  1. Line 71-74: Please rephrase the sentence in order to a better understanding.

Response: According to the reviewer’s comment, we have rewritten this sentence in the revised manuscript (Lines 79-83). Metabolomics analyses were particularly effective in analyzing chemical profiles of fruits from different origins (Lee et al., 2009; Xiang, Wang, Cai, & Zeng, 2011), growing locations (Yang et al., 2012), cultivation ages (Chan et al., 2007) and processing methods (Toh, New, Koh, & Chan, 2010) and played an important role in the variation of chemical characteristics of fruit.

  1. Line 80-83: Give a reference for the stated sentence.

Response: We appreciated for your advices and comments. We have revised the Line 87-90(Metabolomics has also been used to analyze primary metabolites of different citrus varieties, including grapefruit, sweet orange and Persian sour orange, at different fruit picking stages and suggested metabolic pathways associated with citrus fruit development. (Katz et al., 2011; Ledesma-Escobar, Priego-Capote, Robles Olvera, & Luque de Castro, 2018; Lin et al., 2015; Multari, Licciardello, Caruso, & Martens, 2020; Saini, Capalash, Varghese, Kaur, & Singh, 2020).) in the revised manuscript.

Paper is, in general, well write in and experimental parts competently done.  

Response: We appreciated for your advices and comments.

  1. Line 83-86: What analytical technique was used to evaluate these chemical properties and composition?

Response: I apologize for the confusion, in this study, we used the Orthogonal PartialLeast Squares-DiscriminantAnalysis (OPLS-DA) and Principal Component Analysis (PCA) to evaluate these chemical properties and composition.

  1. Line 87-90: Check this sentence, its meaning and sense is not clear.

Response: We appreciated for your advices and comments. We have checked and revised in Line 95-100 (OPLS-DA is widely used in metabolomics research because of its ability to assess similarities and differences between samples and identify key discriminants that lead to differences in variables in the model (Luo et al., 2019; L. Yi et al., 2015). PCA is used to reduce the dimensionality of the data to visualize the overall pattern and clustering of the dataset. OPLS-DA and PCA play crucial roles in synthesizing and interpreting flavonoid data obtained through UPLC-ESI-MS/MS.) in the revised manuscript.

  1. Line 107: What does “every third sample was passed through a 100-mesh sieve” mean? Was not the same procedure for all the samples?

Response: I apologize for the confusion, it was just an inaccuracy in line 108, all the samples was passed through a 100-mesh sieve, we have revised in the revised manuscript.

  1. Line 147: Check the title “statistical analysis”, it is the same than in subsection 2.6. Change.

Response: We thank for the reviewer’s comment. We have checked and revised in the revised manuscript.

  1. Figure 1. (C): If not all the calibration diagrams used for quantitative analysis and, also, the names of the metabolites are given here, this graph does not make so much sense. Add the information or, contrary, move to supplementary material.

Response: We appreciated for your advices and comments. We have moved Figure 1. (C) to supplementary material.

Figure 1. (A)Total Ions Current (TIC) of the mixed-sample. (B) Overlapping of total ions flow diagram of sample.

  1. Figure 2: Give more complete information in the title about the samples analyzed: for example, that they were CRCP collected from July to December… Also, for comparison between samples, it is a key factor of the research process to perform an statistical significance test in order to evaluate if differences found between samples are significant or not. Consider to include it.

Response: We appreciated for your advices and comments. We have revised in the revised manuscript.

Figure 2. The proportion all flavonoid metabolites in CRCP at different maturity periods (July-December)

  1. Table S1: Consider that the total relative content for all the classes of identified compounds would give key information for the study in order to compare the differences between stages.

Response: We thank for the reviewer’s comment. We have moved Table S1 to Table S1 in the revised manuscript.

  1. Line 198: Specify where flavonoids accounted for 36.31%.

Response: We appreciated for your advices and comments. We have revised in Line 212-214 (The flavonoids were analyzed and found to be the most abundant among the identified metabolites in all the samples at different maturity stages, accounting for 36.31% of the total metabolites in all the samples (Jul-Dec).in the revised manuscript.

  1. Line 200-201: It is said that isoflavones, dihydroflavones, dihydroflavonols and flavanols gradually decreased over time but, observing Figure 2 it does not seem to be like that, even showing an increase over time.

Response: We thank for the reviewer’s comment. We have revised in the revised manuscript. Line 215-219 (The relative contents of isoflavones, dihydroflavones and dihydroflavonols gradually increased and flavanols gradually decreased over time. On the other hand, the relative contents of flavonoid C-glycosides showed a pattern of increasing and then decreasing, with the highest content in August.)

  1. Figure 3: As said for Figure 2, give a more detailed title.

Response: We thank for the reviewer’s comment. We have revised in the revised manuscript. Figure 3. Clustering heat map of all flavonoid metabolites in CRCP at different maturity periods (July-December)

Reference:

Fu, M., Xu, Y., Chen, Y., Wu, J., Yu, Y., Zou, B., An, K., & Xiao, G. (2017). Evaluation of bioactive flavonoids and antioxidant activity in Pericarpium Citri Reticulatae (Citrus reticulata ‘Chachi’) during storage. Food Chemistry, 230, 649-656.

Round 2

Reviewer 1 Report

Although the authors have significantly improved the manuscript, two unclear points still remain.

1) Lines 283-284: The concept expressed in the sentence is incorrect. First of all, PCA is an exploratory technique and does not give any proof of reproducibility and reliability. Considering two components is incorrect because the explained variance is low. An additional comment should be made on 3 components, considering the variability explained in this case.

2) Figure 5B: The authors did not respond to a specific observation, namely "Representing the results on a biplot with approximately 35% of the variability is incorrect. And all the remaining 65%? All the considerations made are therefore incorrect." Please, fix this part.

 Finally, a careful revision of the text is suggested because in many places there are -, ., etc. that shouldn't be there.

Author Response

Response to the Reviewer #1

  1. Although the authors have significantly improved the manuscript, two unclear points still remain.

Response: We appreciated for your suggestions and comments.

  1. Lines 283-284: The concept expressed in the sentence is incorrect. First of all, PCA is an exploratory technique and does not give any proof of reproducibility and reliability. Considering two components is incorrect because the explained variance is low. An additional comment should be made on 3 components, considering the variability explained in this case.

Response: Thank you very much for your suggestions and comments and for pointing out the errors. Based on your suggestions, we have revised lines 283-284 in the revised manuscript.

  1. Figure 5B: The authors did not respond to a specific observation, namely "Representing the results on a biplot with approximately 35% of the variability is incorrect. And all the remaining 65%? All the considerations made are therefore incorrect." Please, fix this part.

Response: Thank you very much for your suggestions and comments. The most common use of OPLS-DA analysis results is the OPLS-DA score plot, where the horizontal coordinates indicate the predicted principal components, so that the gaps between groups can be seen horizontally; the vertical coordinates indicate the orthogonal principal components, so that the gaps within groups can be seen vertically; and the percentage indicates how much of the dataset is explained by that component. Each point in the graph represents a sample, samples in the same group are represented by the same color, and groups are subgroups. Then, as the percentages of the horizontal and vertical coordinates cannot be simply added together, in this study, six CRCP samples from different periods were analyzed together for OPLS-DA, and the percentages of the horizontal and vertical coordinates would definitely be lower.

Figure 6. (A) OPLS-DA score 2D graph of samples; (B) OPLS-DA S-plot graph of samples; (C) cross-validation plot of OPLS-DA model with 200 permutation tests.

  1. Finally, a careful revision of the text is suggested because in many places there are -, ., etc. that shouldn't be there.

Response: Thank you very much for your suggestions, we have certainly carefully checked and revised in the revised manuscript.
